# A Rapid and Sensitive Fluorescent Microsphere-Based Lateral Flow Immunoassay for Determination of Aflatoxin B1 in Distillers’ Grains

**DOI:** 10.3390/foods10092109

**Published:** 2021-09-06

**Authors:** Zifei Wang, Pengjie Luo, Baodong Zheng

**Affiliations:** 1College of Food Science, Fujian Agriculture and Forestry University, Fuzhou 350002, China; wangzifei@cfsa.net.cn; 2Chinese Academy of Medical Science Research Unit (No. 2019RU014), NHC Key Laboratory of Food Safety Risk Assessment, China National Center for Food Safety Risk Assessment, Beijing 100022, China; pengjieluo@cfsa.net.cn

**Keywords:** aflatoxin B1, distillers’ grains, fluorescent microspheres, lateral flow immunoassay

## Abstract

Aflatoxin B1 (AFB1) is a toxic compound naturally produced by the genera *Aspergillus*. Distillers’ grains can be used as animal feed since they have high content of crude protein and other nutrients. However, they are easily contaminated by mycotoxins, and currently there are no rapid detection methods for AFB1 in distillers’ grains. In this study, a lateral flow immunoassay (LFIA) based on red fluorescent microsphere (FM), is developed for quantitative detection of AFB1 in distillers’ grains. The whole test can be completed within 15 min, with the cut-off value being 25.0 μg/kg, and the quantitative limit of detection (qLOD) being 3.4 μg/kg. This method represents satisfactory recoveries of 95.2–113.0%, and the coefficients of variation (CVs) are less than 7.0%. Furthermore, this technique is successfully used to analyze AFB1 in real samples, and the results indicates good consistency with that of high-performance liquid chromatography (HPLC). The correlation coefficient is found to be greater than 0.99. The proposed test strip facilitates on-site, cost-effective, and sensitive monitoring of AFB1 in distillers’ grains.

## 1. Introduction

Aflatoxins (AFs), mainly including AFB1, AFB2, AFG1, and AFG2, are a group of secondary metabolites produced by certain fungi (such as *Aspergillus* species). AFs are classified as Group I carcinogens by the International Agency for Research on Cancer [1]. Among them, AFB1 is the most carcinogenic mycotoxin, and has attracted major concerns due to its teratogenic, mutagenic, and carcinogenic effects [2]. The contamination of AFB1 has been frequently found in cereals, oilseeds, spices, tree nuts, and other commodities [3,4,5]. Distillers’ grains are a residual by-product from ethanol industry such as rice, wheat, and sorghum [6]. They have been widely used in animal feed industry since they have a rich source of crude protein and other nutrients [7]. The physicochemical and microbiological characteristics of distillers’ grains have made the product susceptible to mycotoxin contamination [6]. The identification of mycotoxins in distillers’ grains has been reported recently [8,9], raising concern about the potential risk when contaminated distillers’ grains are reused. Moreover, the consumption of contaminated foods and feeds not only poses a severe health threat to both humans and livestock, but also causes non-negligible loss in economy worldwide [10,11]. Many governments and organizations have established the maximum residue limits (MRLs) for AFB1 in different matrices. For example, China and Codex Alimentarius Commission have set MRLs of 20 µg/kg in corn and peanut, and European Union has allowed maximum levels of 5–20 µg/kg for AFB1 in feeds [12]. However, there are currently no MRLs in distillers’ grains. Considering the toxicity and extremely high stability of AFB1, it is imperative to establish a reliable, economical, and high-throughput method to detect AFB1 in distillers’ grains.

Presently, the available methods for detecting AFB1 mainly include instrumental techniques and immunoanalytical approaches. Instrumental methods such as HPLC [13] and liquid chromatography-tandem mass spectrometry (LC-MS) [14,15,16,17] have been commonly used as reference methods due to their high sensitivity. However, they are limited in costly equipment, complex practical procedures, and relatively high requirements for operators. By contrast, the immunoassay such as enzyme-linked immunosorbent assay (ELISA) and LFIA is affordable, rapid, and does not require professionals [18,19], allowing for onsite and reliable analysis [20,21,22]. The lateral flow immunoassay (LFIA), a portable, sensitive, and user-friendly technique, has been developed and evaluated for determining AFB1 in different foods and feedstuffs [23,24]. To effectively improve the sensitivity and selectivity of the immunoassay, different types of signal tracers are involved for qualitative and quantitative detection of AFB1 [25,26,27]. Colloidal gold (CG), a signal marker produced by sub-micrometer nanoparticles of gold, has been widely used in the traditional immunochromatography [10,28,29]. Nevertheless, the application of CG for mycotoxin detection is limited because its sensitivity and stability do not meet certain analytical requirements [11,18,30]. As a result, fluorescent microsphere (FM), a tracer marker with proper brightness, high absorbance, and good stability, has been applied to form the label-antibody conjugate for AFB1 determination [24]. The sufficient amount of carboxyl groups on the surface of this label promotes its binding with the amino groups of proteins, which correspondingly improves the conjugation efficiency and sensitivity of the method [11,18,31,32]. To the best of our knowledge, few studies have focused on the determination of AFB1 in the by-products from ethanol industry such as distillers’ grains via LFIA.

This work aims at establishing and evaluating the LFIA method for detecting AFB1 in distillers’ grain samples. In this study, FM was selected to produce the label-immunoreagent complex; also, an instrumental reader was involved for AFB1 quantification. The preparation for detector reagent (the pH value to active FMs, the amount of Abs, and the optimal coupling buffer) and the performance parameters were optimized to achieve better sensitivity, accuracy, and precision of the method. The proposed technique was applied to detect AFB1 in real samples, and the results showed good consistency with the reference HPLC. This strategy can improve analytical performance for AFB1 determination in complex matrices such as distillers’ grains, and provide advice for better conducting microbial risk assessment when the commodity is consumed.

## 2. Materials and Methods

### 2.1. Reagents and Apparatus

The standard solutions of AFB1, AFB2, AFG1, AFG2, and AFM1 were provided by Pribolab Co., Ltd. (Qingdao, China). Tween-20, 2-(Nmorpholino)-ethanesulfonic acid (MES), 1-ethyl-3-(3-dimethylaminopropyl) carbodiimide hydrochloride (EDC), N-hydroxysuccinimide (NHS), and the goat anti-mouse IgG were purchased from Sigma-Aldrich, Inc. (St. Louis, MO, USA). The bovine serum albumin (BSA) was supplied by Roche Applied Science (Indianapolis, IN, USA). The AFB1-BSA conjugates (5.0 mg/mL) and the AFB1 monoclonal antibody (mAb, 4.0 mg/mL) were produced in our laboratory. The FMs were purchased from Invitrogen (Eugene, OR, USA). The nitrocellulose (NC) membranes, the polyvinyl chloride backing pad, the absorbent pad, and the sample pad was purchased from Jieyi Co., Ltd. (Shanghai, China). The aqueous solution was prepared in ultra-pure water producing by a Milli-Q quality water system (Bedford, MA, USA). The borate buffer solution (BB) was purchased from Aladdin Co., Ltd. (Shanghai, China). All the other chemicals and reagents were of analytical grade and supplied by Aladdin.

The XYZ 3050 Dispensing Platform, and the CM4000 Guillotine Cutter were obtained from Bio Dot, Inc. (Irvine, CA, USA). The transmission electron microscope JEOL JEM-2100 and the fluorescence spectrometer RF-6000 were supplied by Japan Electronics Co., Ltd. (Tokyo, Japan). The results were read by a test reader obtained from Eastwin Co., Ltd. (Suzhou, China), and confirmed by the Agilent 1260 II high performance liquid chromatography system combined with fluorescence detector (HPLC-FLD, Agilent Tech, Santa Clara, CA, USA). The used product model of the fluorescence test reader is eBL-100, and the wavelength is 490 nm. The ZORBAX SB-C18 chromatographic column (5 μm, 4.6 mm × 250 mm, Agilent Technology, Santa Clara, CA, USA) was introduced in this study. The immunoaffinity column (AflaStar R) was obtained from Romer Labs (Beijing, China).

### 2.2. Preparation of FM Immunoprobe

The label–antibody conjugate was prepared via the active ester method [33]. Briefly, 10 μL of FMs were added into 1 mL of 0.05 mol/L MES solution (pH 6.0), then the mixture was stirred in a 1.5 mL centrifuge tube. To active carboxyl groups on the surface of the FMs, 20 μL of EDC (0.5mg/mL) and 24 μL of NHS (0.5mg/mL) were sequentially added and mixed up. After activation for 20 min, the solution was centrifuged at 14,000 rpm for 15 min at 4 °C. The supernatant was discarded and the precipitate was dissolved in 1 mL of 0.5% BSA. Then, 2 μL of AFB1 mAb was added, after incubation for 30 min, 30 μL of 20% BSA was added for 30 min of blocking reaction. Finally, the solution was centrifugated at 14,000 rpm for 15 min at 4 °C. The supernatant was discarded and the residue was reconstituted in 200 μL of resuspension by vortex mixing and sonication. The resuspension was stored at 4 °C for further use.

### 2.3. Fabrication of the Immunochromatographic Strip

The LFIA system consists of a sample pad, an absorbent pad, an NC membrane, and a plastic adhesive backing card. The test strip is performed based on a competitive scheme because the target analyte is a low molecular substance (Figure 1). The AFB1-BSA and the sheep-anti-mouse IgG was diluted to the proper concentrations, filtered, and dispensed at the test line (T line) and control line (C line) at the jetting rate of 0.8 μL/cm respectively. The distance between the T line and the C line was approximately 10 mm. Then, the coated NC membranes were dried at 37 °C overnight. The sample pad was soaked in the 0.02 mol/L phosphate buffer (PB, containing 0.5% BSA, 0.03% Pcoclin-300, 0.3% PVP, and 0.3% Tween-20) for 10 min, and dried at 37 °C for at least 3 h. Lastly, the sample pad, the NC membrane, and the absorbent pad were laminated and coated onto the backing pad. The assembled backing was cut lengthways cut into the width of 3.5 mm and placed in a sealed bag with desiccant.

### 2.4. Sample Pretreatments

The real samples were obtained from the local supermarkets, while the blank samples were confirmed by HPLC. Briefly, the collected distillers’ grain samples were dried in the oven at around 65 °C overnight; then they were ground and properly homogenized for further analysis. The moisture of the samples was determined and the results were correspondingly calibrated based on the statements from ISO 6469. For efficient sample extraction, the representative 1 g sample was mixed with 1 mL of methanol/0.02 mol/L PB (70:30, *v/v*), and vortexed for 5 min for complete extraction. Subsequently, the mixture was centrifugated at 1000 rpm for 10 min. The supernatant was collected and diluted 5 folds with 0.2 mol/L PB (pH 7.4) for analysis.

### 2.5. Immunochromatographic Assay Procedure

The competitive immunoassay format was conducted to detect AFB1 in distillers’ grains. Briefly, 5 μL of FM-AFB1 Ab immunoprobe and 120 μL of the AFB1 standard solution/sample extracts were added into the micropores. Then, they were mixed and gentle pipetted to effectively solubilize the detector particles. After incubation for 3 min, the generated test strip was inserted into the micropores and reacted for another 5 min. After that, the test strip was removed from the micropores and the sample pad was discarded. The results were qualitatively evaluated under the ultraviolet (UV) light. The quantitative analysis was performed by recording the fluorescence intensity at the T line and the C line of the test strip with the reader.

### 2.6. Method Evaluation

#### 2.6.1. Sensitivity

The samples that were free of target analyte were placed into a 15 mL centrifuge tube, and spiked with AFB1 standard solution to achieve the final concentrations of 0, 15, 25, 35, and 45 μg/kg. Then, they were detected according to the aforementioned analytical procedures to estimate the cut-off value, the calibration curve, and the qLOD of the proposed method. Each spiked concentration was analyzed in six times. The cut-off value was calculated as the lowest concentration at which a complete disappearance of fluorescent band at T line was observed. For quantitative evaluation, the calibration curve was constructed by plotting the B/B_0_ percentage values, which is defined as the T/C fluorescence signal ratio in the spiked and blank samples, against the logarithm concentration of AFB1. Specifically, B represents the T/C fluorescence signal in spiked samples, and B_0_ represents the corresponding value in blank samples. Determination of AFB1 levels in real sample was obtained using the calibration curve. The qLOD was calculated as the AFB1 concentration that generated 80% of B/B_0_ value based on the calibration curve [11,18]. The linearity range was defined as the IC_20_–IC_80_ [34].

#### 2.6.2. Specificity

The potential influences from other structural analogs of AFB1 such as AFB2, AFG1, AFG2, and AFM1 was investigated by measuring the cross-reactivity (CR) rate. The CR value was defined as the ratio of IC_50_ (half-maximal inhibitory rate) between target analyte and its analogs.

#### 2.6.3. Accuracy and Precision

The negative control real samples were fortified with AFB1 standard solution at 25, 50, and 100 μg/kg. Then the samples were treated as described above. The recovery and the coefficients of variation (CV) were determined for accuracy and precision evaluation. Moreover, the intra- and inter-day experiments were performed for repeatability and reproducibility studies (*n* = 6).

#### 2.6.4. Confirmatory Test

A total of twenty distillers’ grain samples was collected and analyzed both by the newly established LIFA and the HPLC method coupled with FLD (λex = 360 nm, λem = 440 nm). Each sample was detected in six times. The analytical procedures for sample pre-treatment and the HPLC determination were basing on the AOAC Official Method 2005.08 with some modification. The distillers’ grain samples were purified by the immunoaffinity column (AflaStar R, from Romer Labs Inc., with the column capacity ≥ 200 ng). The instructions from the manufactures were followed. The HPLC was combined with the post-column photochemical derivatization equipment. The mobile phase consisted of water (A) and methanol/acetonitrile (50:50, *v/v*) (B). The isocratic elution was composed of 64% of mobile phase A and 36% of mobile phase B at a flow rate of 1.0 mL/min. The injection volume was 50 μL, and the column temperature was set at 40 °C.

## 3. Results and Discussion

### 3.1. Optimization of the FM Immunoprobe

The immunoprobes used in the LFIA is critical for improving the sensitivity and specificity of the method. The optimal condition in which Ab conjugates with FM needed to be explored [11,32]. The conjugation efficiency was optimized on the aspects of the pH value to activate FMs, the amount of Abs, and the optimal pH value and ion concentration for Ab coupling with FM. The fluorescence intensity, the inhibition effect, and the inhibition rate of the FM immunoprobe were used to interpret the results.

#### 3.1.1. The Activated pH Value of FM

To better conjugate the detector particle with the amino groups of the Ab, the EDC method was used to active the carboxyl groups on the surface of FM [18,32]. The optimal pH value of activation for the conjugation was needed to be optimized [31,35]. The buffer solutions were prepared as following: 0.05 mol/L MES with pH values of 5.0, 5.5, 6.0, and 6.5; 0.01 mol/L PB with pH values of 7.0 and 7.4. The generated FM protein conjugates were dissolved in re-suspension solution and added to the test strips to form the immune complex at the T lines and C lines. As shown in Figure 2a, when the 0.05 mol/L MES (pH 6.0) was applied, the best inhibition rate and satisfactory detectability of fluorescence were achieved. The results suggest that excessively lower or higher pH value can affect conjugation efficiency, and the fluorescence intensity at the T lines will decrease accordingly. Therefore, the optimal pH value was selected as 6.0 when the MES was applied for the conjugation of FMs with the AFB1 Abs.

#### 3.1.2. The Amount of Abs

The amount of Ab applied for the production of FM-immunoreagent was also a key factor for AFB1 analysis in competitive format; indeed, it can affect the conjugation efficiency and the analytical performance [19,29]. Too few antibodies on the surface make the probe’s affinity for the antigen insufficient, and too many antibodies cause protein stacking, which reduces sensitivity [18]. The microspheres (10 μL) and different ratios of AFB1-antibodies (1.0, 1.5, 2.0, and 2.5 μL) were prepared to evaluate the optimal Ab amount for conjugation. In Figure 2b, the fluorescent intensities and the inhibitory rates increased as the amount of Ab rose from 1.0 μL to 1.5 μL, but the color development effect of the test strip was not satisfactory. When the amount of the Abs was 2 μL or more, the test strip could achieve the desired fluorescent intensity. However, less fluorescent signal was observed at T line when 2.5 μL of Ab was used, and the inhibition rate also obviously declined. Taken together, the optimal volume of 2.0 μL of AFB1 Ab was applied in this study.

#### 3.1.3. Selection of the Coupling Buffer

The hydrophobic property of the FM make it easy to agglomerate under certain conditions such as the neutralized charge groups on the microspheres [35]. The combination of Abs and microspheres requires proper pH and ionic strength. The proper pH is usually 6–9. When the ionic strength exceeds 0.2 M, the probe is prone to coagulation [27]. Therefore, the inappropriate ionic strength of the conjugate buffer may lead to agglomeration of the FMs, and in turn render an inefficient conjugation [18,28]. The particles which were used for conjugation should be kept in a stable and dispersed status. To explore the best working condition, six types of different conjugation buffer solutions, including 0.01 mol/L PB (pH 7.4), 0.01 mol/L PB (pH 7.4, 0.5% BSA), 0.05 mol/L BB (pH 8.0), 0.05 mol/L BB (pH 8.0, 0.5% BSA), 0.5% BSA, and ultra-purified water were evaluated (Figure 2c). When 0.5% BSA solution was applied, the brightest fluorescent signal and the best inhibition rate were obtained. Therefore, 0.5% BSA was chosen as the optimal coupling buffer in this study.

### 3.2. Optimization of the Sample Pad Treatment Solution

The sample pad used in the LFIA was to receive the volume of sample and release the target analyte at high performance [36]. Pre-soaking the pads can reduce the matrix effect (ME) and the non-specific binding of the label, and correspondingly improve the sensitivity of the immunoassay system [18,29,32]. Therefore, the formulation of the sample pad treatment solution was investigated and optimized. Generally, the pH value and ionic strength will affect the sensitivity and specificity of the product. The surfactants in the sample pretreatment solution can increase hydrophilicity, have the antigens refolded, and improve the ability of antigen and antibody recognition [11,27]. Consequently, the factors such as the surfactant concentration, the ionic strength, and the pH value were taken into consideration. Here, the optimized pretreatment solution consists of 0.5% BSA, 0.03% Proclin-300, 0.3% PVP, 0.3% Tween-20, all added into 0.02 mol/L PB (pH 7.4). The effect of Tween-20, which served as a surfactant in the sample soaking solution [25,37], was also studied for enhancing the color development and inhibition rate. The evaluation results (Figure 3a) indicated that when the concentrations of Tween-20 rose from 0.1% to 0.3%, a good brightness of the fluorescent signal was generated, accompanied with the improvement of the inhibition rate. However, both the label intensity and the inhibition rate were decreased to different levels when the concentrations of Tween-20 subsequently increased from 0.3% to 1.3%. To further explore the effects of the ionic strength and the pH value, four kinds of sample pad treatment buffer were tested and evaluated (Figure 3b). Among the analyzed buffer solutions, the most satisfactory fluorescent intensity was achieved with the usage of 0.02 mol/L PB (pH 7.4). Under this optimized condition, the inhibition was also achieved to the highest level. Taken together, 0.02 mol/L PB (pH 7.4) with 0.3% Tween-20 was used to improve the analytical performance.

### 3.3. Evaluation of the Test Strip

#### 3.3.1. Sensitivity

The sensitivity of the method was evaluated by measuring AFB1 concentrations in a series of spiked samples. In the competitive format, the fluorescent intensity at the T line is inversely related to AFB1 content in the sample. The results can be read both by the naked eyes and the test strip reader (Figure 4). The cut-off values were 25 μg/kg in all the samples, when the fluorescent signal of the T line completely disappeared. According to the established calibration curve, the qLOD was found to be 3.4 μg/kg for distillers’ grains, and the detected range (IC_20_–IC_80_) was from 3.4 to 34.7 μg/kg.

#### 3.3.2. Selectivity

To test the selectivity of the method, the analogs of AFB1 such as AFB2, AFG1, AFG2, and AFM1 were also analyzed in spiked real samples. According to the results (Table 1), the cross reactivity (CR) was negligible to AFG1, AFG2 and AFM1 (<0.05%), but relatively apparent for AFB2. The results were possibly due to the higher similarity of structure and molecular constitution between AFB1 and AFB2 when compared with other analogs.

#### 3.3.3. Accuracy and Precision

The samples free of target compound were spiked with AFB1 standard solution at 5, 15, and 30 μg/kg. According to the results listed in Table 2, the recovery rates ranged from 95.2% to 113.0%, with all the CVs less than 7% (*n* = 6). For precision investigation, the intra-day repeatability exhibited recovery rates from 89.0% to 106.7%, with the CVs < 7.0%; the inter-day assay also exhibited satisfactory reproducible results (the recovery rates of 93.0% to 106.8%, the CVs of 1.3% to 4.4%) (Table 3).

### 3.4. A Comparison between the Test Strip and HPLC

To further validate the method, twenty real samples were randomly collected and simultaneously detected by our established FM-LFIA and the confirmatory method, HPLC [32,35] (Table 4). The results obtained by the LFIA were in good consistency with that of HPLC (*r* > 0.99). Among the analyzed samples, six sample was detected to be positive, and the results were also validated by HPLC. The results suggest that the proposed technique is applicable for the on-site quantitative detection of AFB1, and facilitates the rapid screening of samples.

### 3.5. Comparison of Different Immunoassays for AFB1 Detection

The comparison of different immunoassays for AFB1 determination in food samples are summarized in Table 5. The results indicate that the food matrices involved in previous studies typically limit to cereal, corn, and feedstuff originated from corn. Although some techniques exhibit higher sensitivity than our proposed method, the relatively high cost and complex operational procedures should be taken into consideration when the routine analysis is conducted. Presently, the frequent occurrence of mycotoxins in distillers’ grains has attracted increasing concerns around the world [38], but there is no data on the determination of AFB1 in this by-product via LFIA. Therefore, this work has developed a rapid and sensitive LFIA for detecting AFB1 in distillers’ grains, which could meet the MRL of 20 µg/kg in corn and peanut set by China and Codex Alimentarius Commission.

## 4. Conclusions

In this study, a sensitive, economical, and rapid immunoassay was established for AFB1 determination in distillers’ grain samples. The FM was employed to improve the sensitivity of the technique. The cut-off value was 25 μg/kg in distillers’ grain samples, and the qLOD was 3.4 μg/kg. The recovery rates ranged from 95.2% to 113.0%, with all the coefficient of variations (CVs) less than 7% (*n* = 6). The proposed method represents satisfactory analytical parameters, such as sensitivity, selectivity, and precision. To further evaluate the feasibility of the immunoassay, twenty real samples were tested by the proposed method and the referenced HPLC-FLD; strong agreement was observed when the results were interpreted. This strategy can improve analytical performance for AFB1 determination, and provide advice for better conducting toxicological risk assessment in distillers’ grains.

## Figures and Tables

**Figure 1 foods-10-02109-f001:**
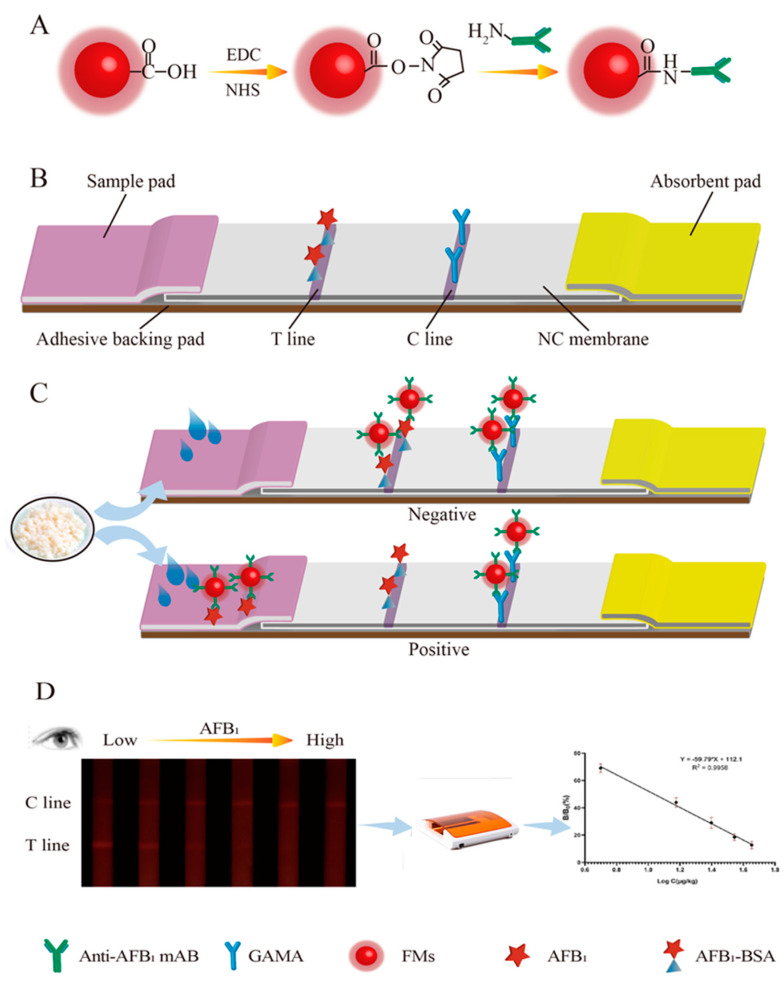
Schematic diagram of FMs-LFA for AFB_1_ detection in distillers’ grains: (**A**) the preparation principle of FMs-AFB_1_ Ab immunoprobe; (**B**) the structure of test strip; (**C**) the detection principle of test strip; and (**D**) the qualitative and quantitative test results.

**Figure 2 foods-10-02109-f002:**
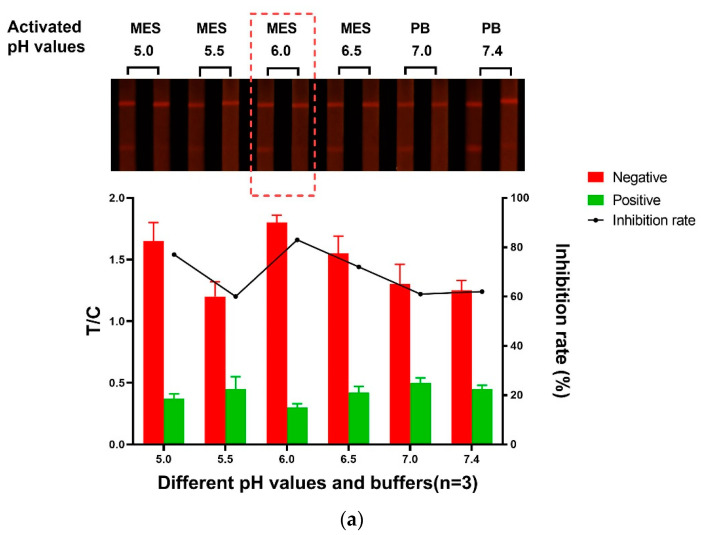
The fluorescence intensity, inhibition effect, and inhibition rate of the FM immunoprobe: (**a**) Activation pH value; (**b**) Amount of the AFB1 monoclonal antibody; (**c**) the conjugation buffer.

**Figure 3 foods-10-02109-f003:**
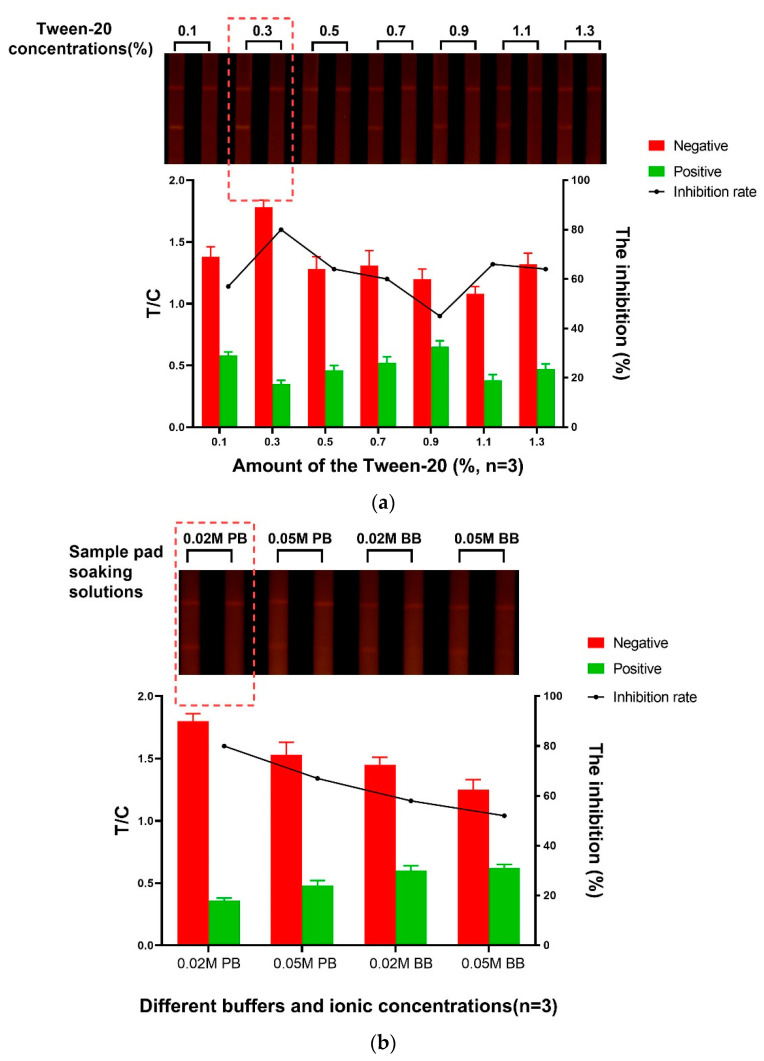
The fluorescence intensity, inhibition effect, and inhibition rate of the Sample pad treatment: (**a**) Tween-20 concentration; (**b**) The different sample pad soaking solutions.

**Figure 4 foods-10-02109-f004:**
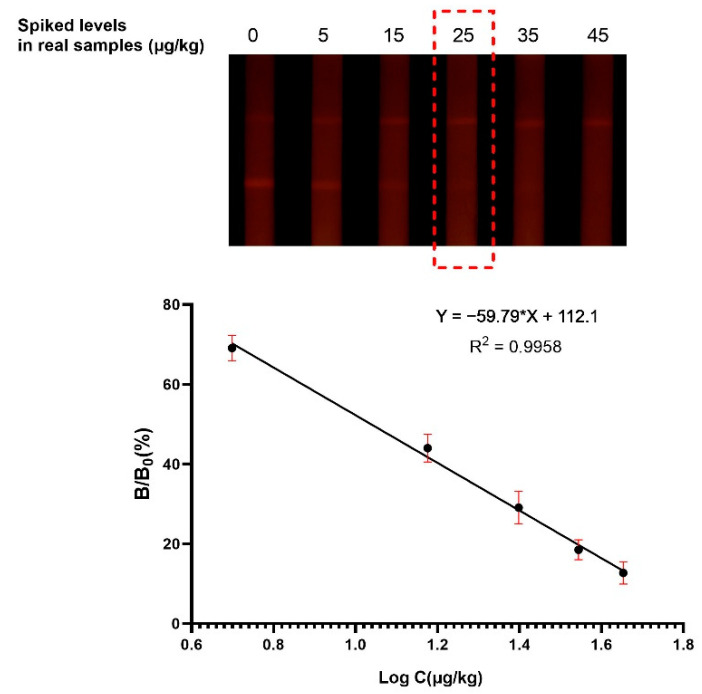
The images and the calibration curve of AFB1 in distillers’ grain samples. The cut-off value was 25 μg/kg, and the qLOD was 3.4 μg/kg.

**Table 1 foods-10-02109-t001:** The cross reactivity of AFB1 against other analogues (AFB2, AFG1, AFG2, AFM1).

Mycotoxins	Structure	CR (%)
AFB1	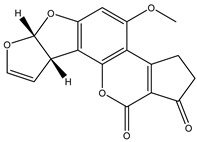	100
AFB2	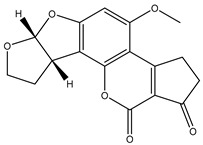	5.2
AFG1	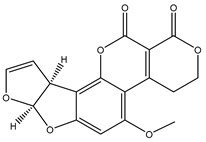	0.02
AFG2	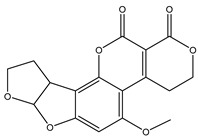	0.01
AFM1	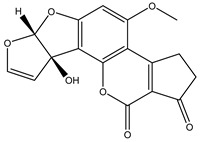	0.04

**Table 2 foods-10-02109-t002:** The recovery rates of LFIA for AFB1 determination (*n* = 6).

Spiked Levels (μg/kg)	Mean ± SD (μg/kg)	Recovery (%)	CV (%)
5	5.3 ± 0.3	95.2–113.0	6.1
15	14.8 ± 0.5	96.1–104.3	3.1
30	31.3 ± 0.7	102.0–108.5	2.1

**Table 3 foods-10-02109-t003:** Intra- and inter-day studies of AFB1 determination in distillers’ grain samples.

Spiked Levels (μg/kg)	Intra-Day Precision, *n* = 6		Inter-Day Precision, *n* = 6	
	Recovery (%)	CV (%)	Recovery (%)	CV (%)
5	89.0–106.5	6.5	93.0–105.2	1.3
15	90.5–103.5	5.1	96.2–103.1	1.4
30	96.7–106.7	3.8	96.6–106.8	4.4

**Table 4 foods-10-02109-t004:** Determination of AFB1 in real samples by LFIA and HPLC methods.

Samples	AFB1 Concentration (μg/kg)	
	LFIA	HPLC
Sample 1	N.D. ^1^	N.D.
Sample 2	6.14	5.95
Sample 3	N.D.	N.D.
Sample 4	7.45	7.81
Sample 5	N.D.	N.D.
Sample 6	N.D.	N.D.
Sample 7	8.21	8.13
Sample 8	N.D.	N.D.
Sample 9	N.D.	N.D.
Sample 10	N.D.	N.D.
Sample 11	5.82	5.69
Sample 12	N.D.	N.D.
Sample 13	N.D.	N.D.
Sample 14	N.D.	N.D.
Sample 15	N.D.	N.D.
Sample 16	N.D.	N.D.
Sample 17	7.21	7.38
Sample 18	N.D.	N.D.
Sample 19	4.95	5.12
Sample 20	N.D.	N.D.

^1^ N.D. indicated that the analyzed AFB1 concentrations were lower than the qLOD of the method.

**Table 5 foods-10-02109-t005:** Comparison of different immunoassays for AFB1 determination in foods.

Method	Sample	Sensitivity	Reference
Smartphone-Based LFIA	Maize	The detection limit for AFB1 was 5 μg/kg	[39]
ELISA	Corn	The limit of detection (LOD) for AFB1 was 12 ± 2.3 ng/mL	[17]
Au-particles-based LFIA	Corn	The visual LOD for AFB1 was 10 ng/mL	[40]
Quantum-dot based fluorescence immunoassay	Cereal	The LOD for AFB1 was 0.01 ng/mL	[25]
ELISA test kit	Corn feed	The LOD for AFB1 was 1.1 μg/kg and limit of quantification was 2.5 μg/kg	[22]
FM-based LFIA	Distillers’ grain	The cut-off value for AFB1 was 25.0 μg/kg, and the qLOD was 3.4 μg/kg	This work

## Data Availability

The datasets generated for this study are available on request to the corresponding author.

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
