# Peer review of "A Rapid and Sensitive Fluorescent Microsphere-Based Lateral Flow Immunoassay for Determination of Aflatoxin B1 in Distillers’ Grains"

_foods, 2021, doi:10.3390/foods10092109_

Round 1

Reviewer 1 Report

Title:  A rapid and sensitive fluorescent microsphere-based lateral flow immunoassay for determination of Aflatoxin B1 in distillers’ grains.

General Comments:

The authors took an existing lateral flow immunoassay and optimized it for potential use in detecting AFB1 in distillers' grains.  The idea was worth pursuing and their results illustrate the potential of the modified method for the proposed application.  Also, the method could be of significance to the associated industries.  However, the authors also suggested that the resulting method has application for high sample throughput.  Considering the study only tested 6 real world distillers' grain samples, such a statement is overreaching.  Many more samples would need to be tested to validate the method and its potential use for high volume sampling. 

Overall impression of the manuscript indicates a need for extensive editing of the English language, stylistically and grammatically.  Additional suggestions are outlined below.

Specific suggestions:

The introduction generally ends with a paragraph describing the specific aims/objectives for the study.  What is being done? Why? How? This is not clear in the write-up.

The results and discussion section needs improvement.  Some of the methodological details in the results belong in the methods section.  It also needs to be explained why certain buffers, etc., were chosen for optimizing.  The discussion needs to be expanded and the results need to be explained /interpreted in context of existing literature.  Please refer to references #25 and 27 in your reference section for ideas of how to expand the result/discussion section.  Number 27...Chen et al 2016, in particular, is very helpful. 

Author Response

Foods Title: A rapid and sensitive fluorescent microsphere-based lateral flow immunoassay for determination of Aflatoxin B1 in distillers’ grainsManuscript No.: 1328260

Responses to reviewers' comments

Reviewer: #1

General Comments:

  1. The authors took an existing lateral flow immunoassay and optimized it for potential use in detecting AFB1 in distillers' grains. The idea was worth pursuing and their results illustrate the potential of the modified method for the proposed application. Also, the method could be of significance to the associated industries. However, the authors also suggested that the resulting method has application for high sample throughput. Considering the study only tested 6 real world distillers' grain samples, such a statement is overreaching. Many more samples would need to be tested to validate the method and its potential use for high volume sampling.

Response:Thanks for reviewer’s professional guidance. We analyzed more real samples and supplemented the relevant data in section 3.4 and Table 4. Besides, the whole test can be finished within 15 min, which facilitates its practical application for high throughput samples.

  1. Overall impression of the manuscript indicates a need for extensive editing of the English language, stylistically and grammatically. Additional suggestions are outlined below.

Response:Thanks for reviewer’s kindly suggestion. The entire manuscript has been polished by a native speaker, all changes in the revised manuscript have been marked in red.

Specific suggestions:

1.The introduction generally ends with a paragraph describing the specific aims/objectives for the study.  What is being done? Why? How? This is not clear in the write-up.

Response: Thanks for reviewer’s professional guidance. The last paragraph of the introduction has been revised as suggested. For the revised paragraph, please see lines 67-72 in the revised manuscript.

2.The results and discussion section needs improvement. Some of the methodological details in the results belong in the methods section. It also needs to be explained why certain buffers, etc., were chosen for optimizing. The discussion needs to be expanded and the results need to be explained /interpreted in context of existing literature.  Please refer to references #25 and 27 in your reference section for ideas of how to expand the result/discussion section. Number 27...Chen et al 2016, in particular, is very helpful.

Response: We appreciate reviewer’s suggestion. The results and discussion section have been improved, some method optimizations have been explained, and the scope of discussion has also been expanded. For the section 3.1.2, “Too few antibodies on the surface make the probe's affinity for the antigen insufficient, and too many antibodies cause protein stacking, which reduces sensitivity” was added in lines 221-223 in the revised manuscript. For the section 3.1.3, “The combination of Abs and microspheres requires proper pH and ionic strength. The proper pH is usually 6–9. When the ionic strength exceeds 0.2 M, the probe is prone to coagulation” was added in lines 234-236 in the revised manuscript. For the section 3.2, “Generally, the pH value and ionic strength will affect the sensitivity and specificity of the product. The surfactants in the sample pretreatment solution can increase hydrophilicity, have the effect of refolding antigens, and improve the ability of antigen and antibody recognition” was added in lines 258-261 in the revised manuscript.

Reviewer 2 Report

This manuscript describes a lateral flow immunoassay based on red a fluorescent microsphere for quantitative detection of AFB1 in distillers’ grains.

The main idea is a well-known method. There have already been similar reports for the detection of AFB1 by LFIA. More importantly, the main concept of this manuscript is almost similar to the one published in Food control in 2016 and Toxicon in 2018. Therefore, I recommend major revision.

Zhang, Xiya, et al. "An ultra-sensitive monoclonal antibody-based fluorescent microsphere immunochromatographic test strip assay for detecting aflatoxin M1 in milk." Food Control 60 (2016): 588-595.

Yu, Songcheng, et al. "A lateral flow assay for simultaneous detection of deoxynivalenol, fumonisin B1 and aflatoxin B1." Toxicon 156 (2018): 23-27.

  1. The authors should provide a table comparing this method with other previous methods for the detection of Aflatoxin B1 (AFB1).
  2. Lateral flow immunoassays for detection of Aflatoxin in grains have already been reported. Is there a difference in the detection of aflatoxin in grain and distillers’ grains?
  3. In this method, AFB-BSA is pre-attached to the C line and then the target AFB sample is put into the sample pad. The difference in competitive binding between AFB and mAb-FM was detected. This is an inefficient detection method because it cannot directly recognize the target and also represents off-signal results. The authors should accurately describe the novelty of the technology.
  4. The proposed test strip requires additional fluorescence detector compared to the LFA using gold nanoparticles. What are the advantages compared to colorimetric LFA system?
  5. Please check the spelling of line 15.
  6. There are no error bars in all experimental data except for sensitivity experiments. It is recommended to write an error bar on the image and write the number of times the experiment was repeated.
  7. In figures 2(c) and 3(b), 0.01 mol/L PB is indicated as 0.01 MPB, which may cause misunderstanding. It is recommended to write 0.01M PB by inserting a space.
  8. According to the reference 23, there are a lot of technologies with better sensitivity of less than 1 ug/kg. What is the reason for the low sensitivity of author’s method compared to the relevant technologies and what are the advantages of this technology?

Author Response

Foods Title: A rapid and sensitive fluorescent microsphere-based lateral flow immunoassay for determination of Aflatoxin B1 in distillers’ grainsManuscript No.: 1328260

Responses to reviewers' comments

Reviewer: #2

  1. The authors should provide a table comparing this method with other previous methods for the detection of Aflatoxin B1 (AFB1).

Response: Thanks for your good suggestion. We have supplemented “Table 5” and “section 3.5” for comparison between this method and other immunoassays for the detection of AFB1. The section 3.5 are descripted as follows: The comparison of different immunoassays for AFB1 determination in food samples are summarized in Table 5. The results indicate that the food matrices involved in previous studies typically limit to cereal, corn, and feedstuff originated from corn. Although some techniques exhibit higher sensitivity than our proposed method, the relatively high cost and complex operational procedures should be taken into consideration when the routine analysis is conducted. Presently, the frequent occurrence of mycotoxins in distillers’ grains has attracted increasing concerns around the world, but there is no data on the determination of AFB1 in this by-product via LFIA. Therefore, this work has developed a rapid and sensitive LFIA for detecting AFB1 in distillers’ grains, which could meet the MRL of 20 µg/kg in corn and peanut set by China and Codex Alimentarius Commission.

  1. Lateral flow immunoassays for detection of Aflatoxin in grains have already been reported. Is there a difference in the detection of aflatoxin in grain and distillers’ grains?

Response: Distiller's grains are the residual left-over from ethanol industry such as rice, wheat, sorghum, etc. Distillers' grains can be used as the ideal animal feed since they have high content of crude protein and other nutrients. This by-product is rich in crude protein, which is about 2-3 times higher than that of corn. It also contains a variety of trace elements, vitamins, yeasts, etc. Besides, the contents of lysine, methionine and tryptophan in distillers’ grains are also very high. It is a more ideal animal feed than grain. However, compared with grains, distiller's grains are more vulnerable to biological toxin pollution, and there is a lack of rapid and onsite detection methods for distiller's grains at present.

  1. In this method, AFB-BSA is pre-attached to the C line and then the target AFB sample is put into the sample pad. The difference in competitive binding between AFB and mAb-FM was detected. This is an inefficient detection method because it cannot directly recognize the target and also represents off-signal results. The authors should accurately describe the novelty of the technology.

Response:Thanks for reviewer’s professional guidance. We are very sorry that there were errors in the detection schematic diagram (Figure 1) and it has been revised, please check it. The proposed technology is based on the specific combination of antigen and antibody, which not only possesses the fast speed of homogeneous immunoassay, but also keeps the advantage of heterogeneous methods which can efficiently separate reacted and unreacted compounds. Fluorescent microspheres (FMs), the special polystyrene beads containing fluorescent substance in their interior, have become a new promising type of fluorescence labels. FMs and antibodies are covalently bound, and they are not easily affected by high protein, high fat, high salt components and strong acids and bases in complex matrices, resulting in antibody shedding or probe aggregation, which improves the stability and reproducibility of immunochromatography. In general, FMs-LFA has the characteristics of strong anti-matrix interference and high colorimetric signal intensity. Besides, this newly developed technique is particularly suitable for onsite analysis in the workplaces for Chinese liquor production, which can effectively control AFB1 levels in ethanol industry.

  1. The proposed test strip requires additional fluorescence detector compared to the LFA using gold nanoparticles. What are the advantages compared to colorimetric LFA system?

Response: Compared to colorimetric LFA system, the greatest advantage is the replacement of antibody labeled tracers. Both colloidal gold and fluorescent microspheres are antibody labeled tracers. Colloidal gold (CG) is the most widely used labels because of their simple and rapid synthesis, low cost, and easy interpretation of obtained analytical results. However, the low bright intensity of conventional CG results in poor LFA sensitivity. Fluorescent microspheres (FMs) have been commonly used in immunoassays because of their stable morphology, high fluorescence efficiency, and the ability of surface modification. The surface of the FMs is modified with a certain density of carboxyl or other functional groups, and covalently coupled with the proteins or antibodies to greatly improve the stability and sensitivity of the method. Moreover, the fluorescence reader can used to quantify the results and further improve the sensitivity of the method.

  1. Please check the spelling of line 15.

Response: Thank you for your careful review. we have checked the spelling of line 15 and corrected it.

  1. There are no error bars in all experimental data except for sensitivity experiments. It is recommended to write an error bar on the image and write the number of times the experiment was repeated.

Response: We appreciate reviewer’s suggestion. We had supplemented the error bars and the number of repeated experimental times in Figure 2 and Figure 3.

  1. In figures 2(c) and 3(b), 0.01 mol/L PB is indicated as 0.01 MPB, which may cause misunderstanding. It is recommended to write 0.01M PB by inserting a space.

Response: Thanks for your careful review. We have clarified the description in the Figure 2(c) and 3(b).

  1. According to the reference 23, there are a lot of technologies with better sensitivity of less than 1 ug/kg. What is the reason for the low sensitivity of author’s method compared to the relevant technologies and what are the advantages of this technology?

Response: Compared to the previous reported technologies, the low sensitivity is mainly due to the detected samples. In this study, the tested sample was the distiller's grains. Distiller’s grain is regarded as a type of complex matrix sample that is difficult to handle in the immunochromatographic technology industry. Compared with grains, the high content of protein and cellulose in the matrices of distiller's grains can easily enhance the matrix effects on the test results, and directly affect the performance of the immunochromatography. In order to reduce the matrix effect, the sample needs to be diluted, and the dilution factor will affect the sensitivity of the method.

The advantage of our technology is that FMs were employed to as the antibody labeled tracers. The surface of the FMs is modified with a certain density of carboxyl or other functional groups, and covalently coupled with the proteins or antibodies to greatly improve the stability and sensitivity of the LFIA. Moreover, the synthesis of FMs has also been commercialized to ensure the reproducibility and applicability of the method. Once this technology is mature, it can be marketed soon.

Moreover, this newly developed technique is especially suitable for onsite analysis in the workplaces for ethanol production in China, which can take proper control of AFB1 levels in ethanol industry.

Round 2

Reviewer 1 Report

The authors addressed some of the reviewer's concerns, however the manuscript still needs work.  The introduction must be reworked and the English language/style needs extensive editing.

If the authors would like assistance with these tasks, please submit a word document so changes can be tracked.  Otherwise, line by line editing is cumbersome.

Author Response

Thanks for your careful review! We have carefully revised the whole manuscript according to the comments. Specifically, the introduction has been reworked and the revised statements have been marked in red. The English language/style has also been checked and edited.

Reviewer 2 Report

I think the manuscript is now acceptable in Foods

Author Response

Thanks for your kind consideration!